# Design, Synthesis and Antifungal Evaluation of Novel Pyrylium Salt In Vitro and In Vivo

**DOI:** 10.3390/molecules27144450

**Published:** 2022-07-12

**Authors:** Yue Zhang, Qiuhao Li, Wen Chao, Yulin Qin, Jiayan Chen, Yingwen Wang, Runhui Liu, Quanzhen Lv, Jinxin Wang

**Affiliations:** 1School of Pharmacy, Naval Medical University, Shanghai 200433, China; yzhang_moon@163.com (Y.Z.); lqh20191034@163.com (Q.L.); chenjiayan_alice@outlook.com (J.C.); wyw20010625@163.com (Y.W.); 2Experimental Teaching Center of Basic Medicine College, Navel Medical University, Shanghai 200433, China; chaowen_2010@163.com; 3Fudan University Minhang Hospital, Shanghai 201199, China; qinyulin1990@126.com

**Keywords:** pyrylium salt, SM21, antifungal, *Candida albicans*, drug discovery

## Abstract

Nowadays, discovering new skeleton antifungal drugs is the direct way to address clinical fungal infections. Pyrylium salt **SM21** was screened from a library containing 50,240 small molecules. Several studies about the antifungal activity and mechanism of **SM21** have been reported, but the structure–activity relationship of pyrylium salts was not clear. To explore the chemical space of antifungal pyrylium salt **SM21**, a series of pyrylium salt derivatives were designed and synthesized. Their antifungal activity and structure-activity relationships (SAR) were investigated. Compared with **SM21**, most of the synthesized compounds exhibited equivalent or improved antifungal activities against *Candida albicans* in vitro. The synthesized compounds, such as **XY10**, **XY13**, **XY14**, **XY16** and **XY17** exhibited comparable antifungal activities against *C. albicans* with MIC values ranging from 0.47 to 1.0 μM. Fortunately, a compound numbered **XY12** showed stronger antifungal activities and lower cytotoxicity was obtained. The MIC of compound **X****Y12** against *C**. albicans* was 0.24 μM, and the cytotoxicity decreased 20-fold as compared to **SM21**. In addition, **XY12** was effective against fluconazole-resistant *C. albicans* and other pathogenic *Candida* species. More importantly, **XY12** could significantly increase the survival rate of mice with a systemic *C. albicans* infection, which suggested the good antifungal activities of **XY12** in vitro and in vivo. Our results indicated that structural modification of pyrylium salts could lead to the discovery of new antifungal drugs.

## 1. Introduction

*Candida*, *Aspergillus* and *Cryptococcus* are the main opportunistic pathogens leading to human fungal infections [1]. During the last three decades, invasive fungal infections (IFIs) have killed about 1.6 million people each year and pose a serious threat to human health, especially among people with HIV infection, cancer, organ transplants and autoimmune diseases [2]. In recent years, systemic *Candida* infection has risen to fourth place among the nosocomial blood-borne infections [3,4]. At the same time, the mortality rate of candidemia is high. According to the statistics, the 90-day mortality rate of solid-organ transplant recipients with candidiasis is 22–44% [5]. These studies indicate that pathogenic fungi are a serious and increasing threat to human health. Solving the problem of fungal infections is an important means to improve the recovery rate of patients with immunodeficiencies [6].

Despite advances in antifungal therapy, invasive fungal infection remains the major cause of morbidity and death among immunocompromised patients [7]. Clinically, three major classes of drugs are available, which include azoles, echinocandins and polyenes, but the mortality of patients with invasive fungal infection is still as high as 40% [8,9]. In addition, the long-term use of antifungal drugs causes the problem of drug resistance. Multi-center investigations of candidiasis in China showed that about 0.5–2.0% of *C. albicans* was resistant to fluconazole. Moreover, the survival time of patients infected with drug-resistant *C. albicans* was significantly lower than that of patients infected with sensitive strains [10]. With continuous emergence of drug resistance and the undesired side effects of available antifungal drugs, effective agents with new structures and targets are urgently needed.

In 2014, Wong et al screened 50,240 compounds for their antifungal activities using the yeast-to-hyphae inhibitory phenotype. Among them, lipophilic pyrylium salt **SM21** (Figure 1
**XY2** in this study) showed good antifungal activities in vitro and in vivo [11]. Related mechanistic studies demonstrate that **SM21** could inhibit the function and fusion of mitochondria in *C. albicans* [12,13]. However, the structure–activity relationship of **SM21** is not clear. In our studies, a series of pyrylium salts were rationally designed and synthesized base on the chemical space investigation of **XY2** (Figure 1). The structure–activity relationship of pyrylium salts was preliminarily discussed and a compound numbered **XY12** was obtained, which showed stronger antifungal activity and lower cytotoxicity. Our research provided a new lead compound for the development of antifungal drugs.

## 2. Results and Discussion

### 2.1. Chemistry

Except for commercially available pyrylium salt derivatives, new pyrylium salt derivatives **XY3**-**20** were synthesized from **XY1A** or **XY1B** (Figure 1). The synthesis of the target compounds **XY3-17**, **XY19** and **XY20** was shown in Figure 1, Figure 2 and Figure 3, respectively. Pyrylium salts were used as cationic dyes [14] and the methods to prepare these types of compounds were reported [15]. The target compounds **XY3-17** and **XY19** can be obtained efficiently by a dehydration condensation reaction of methyl pyrylium salt **XY1A** or **XY1B** and a different substituted aldehyde **A1-16** under the heating condition in acetic anhydride, with single trans-olefin selectivity (Figure 1 and Figure 2). The structures of new pyrylium salts were determined by the high resolution mass spectra and ^1^H, ^19^F and ^13^C-NMR spectra shown in the Appendix A. All the synthesized compounds contained triflate (CF_3_SO_3_^−^) as the counter anions were established from the HRESIMS data in negative ion mode and the peak of −79 ppm in the ^19^F-NMR spectra. The ^1^H-NMR signals at *δ*_H_ 1.40 (s, 9 H) and *δ*_H_ 7.64 (s, 2 H) could be attributed to the pyrylium motif. The trans-olefin was confirmed by the doublet olefinic proton signals with a coupling constant of 15.2 Hz.

Moreover, the neutral analogue **XY20** was prepared by the Horner-Wadsworth-Emmons reaction. The treatment of aldehyde **A11** with dibenzyl phosphate **S1** under alkaline condition afforded *E*-type olefin **XY20** in a 94% yield (Figure 3).

### 2.2. Structure-Activity Relationship (SAR) Studies

The main purpose of this study is to explore the chemical space and analyze the structure-activity relationship of pyrylium salt derivatives against fungi to find novel skeleton antimicrobial agents with high efficiency and low toxicity. A series of pyrylium salt derivatives were designed and synthesized and their antifungal activity was evaluated in vitro by standard pathogenic fungi *C. albicans* SC5314. These results are summarized in Table 1.

Firstly, the structure of **XY2** (**SM21**) was analyzed. Pyrylium salt (**XY1A**) and 4-(Dimethylamino)benzaldehyde (**A1**) showed no significant activity (MIC > 179.7 μM), which revealed that both fragments of the motif were required for antifungal activity. Pyrylium salt is a kind of cationic dye [14], in which the terminal groups have a dramatic effect on their properties, such as the properties of electron transition and spectral properties. Based on these properties, a series of pyrylium salt compounds (**XY2-17**) with different substituents were synthesized. Interestingly, a formal positive correlation between antifungal activity and spectral properties was observed. It is to be noted that the electron-donating levels, generated predominantly by terminal groups, certainly determined the antifungal activity. Compounds **XY7**, **XY8** and **XY9,** bearing a strong electron-withdrawing group, had no antifungal activity. At the same time, pyrylium salt derivatives without a substituent (**XY5**) or with a fluorine substituent (**XY6**) showed low activity, and the MIC was higher as 138.5 μM. As expected, the antifungal activity of methoxyl **XY3** (MIC = 67.5 μM) and phenolic hydroxyl **XY4** (MIC = 34.8 μM) were also much weaker than **XY2**. The above results prompted us to further study the substituent groups on nitrogen. By replacing dimethylamine (**XY2**) with diethylamine (**XY10**) and diphenylamine (**XY11**), the results showed the activity of diphenylamine decreased significantly and diethylamine (MIC = 0.49 μM) was equal to dimethylamine (MIC = 0.51 μM). Taken together, these results indicate that *N*,*N*-dialkylaniline as the terminal group is a necessary, but not sufficient, prerequisite for antifungal activity.

Next, a series of cyclic alkylamine derivatives were designed and synthesized for the antifungal evaluation (**XY12-15**). To our surprise, ring size was directly associated with the antifungal activity, with the five-membered ring being the most potent antifungal agent with an MIC value of 0.24 μM (**XY12**), whereas the activity of both four-membered and six-membered rings decreased (**XY13**-**14**, MIC = 0.95–1.0 μM). In addition, the introduction of a further heteroatoms such as morpholine, resulted in the decline in the activity (**XY15**, MIC = 3.8 μM). On this basis, the introduction of the fluoro-substituted and hydroxyl-substituted derivatives (**XY16**-**17**) exhibited decreased activity (0.47 μM, 0.95 μM). It is possible that steric hindrance of substituents affects the activity of compounds. Thus far, we obtained the best compound, **XY12**.

Finally, to evaluate the effect of substitutions on the pyrylium ring, the dicyclopentyl-substituted derivative **XY18** (commercially available) and diphenyl-substituted derivative **XY19** were synthesized, starting with the methyl pyrylium salt (**XY1B**) and aldehyde **A11** using the standard synthetic route described above (Figure 2); they exhibited a decline in antifungal potency (8.7 and 3.6 μM, respectively) compared to *tert*-butyl-substituted counterparts (**XY2**, **XY12**). **XY20** was obtained by the Horner–Wadsworth–Emmons reaction (Figure 3), and its pyrylium ring was replaced by a benzene ring. The inactivation of **XY20** (>177.1 μM) indicated the necessity of the cation (Table 1). This result further implied that this lipophilic pyrylium dye shows antifungal activity, probably via targeting mitochondria [12,13,16,17].

### 2.3. Pharmacological Activity of ***XY12***

To further investigate the antifungal activity of **XY12**, a variety of *Candida* strains were used to test the MIC of **XY12** and **XY2**. Compared with the antifungal activity of **XY2**, the MIC of **XY12** decreased about one-fold. **XY12** showed enhanced antifungal bioactivity against most of the tested *Candida* species with the MIC ranging from 0.12 μM to 0.97 μM, including *C. albicans*, *C. glabrata*, *C. tropicalis*, *C. krusei* and *C. parapsilosis* (Table 2). The compound **XY12** also showed excellent activity against fluconazole-resistant *C. albicans* 100 and 901 with the MIC value of 0.24 μM and 0.12 μM (Table 2). By comparing the radar charts, we clearly found that the MICs of **XY12** against the pathogenic *Candida* were lower than those of **XY2** (Figure 2), which indicated the stronger anti-*Candida* activities of **XY12**.

To investigate the cytotoxicity of **XY12** and **XY2**, the viability of human umbilical vein endothelial cells (HUVECs) was assayed by CCK-8 agents. As shown in Figure 3, 7.8 μM of **XY12** showed almost no toxicity to HUVECs and the IC_50_ of **XY12** was about 34.6 μM. However, a significant inhibitory effect of **XY2** was observed at the concentration of 0.51 μM. Meanwhile, the IC_50_ of **XY2** was about 1.8 μM, which indicated that the cytotoxicity of **XY12** was reduced by about 20-fold. Therefore, structural modification of **XY2** at the terminal groups not only improves the antifungal activity, but also reduces the cytotoxicity.

Finally, we compared the in vivo antifungal activity of **XY12** and **XY2** using the systemic candidiasis model. Treated with 10 mg/kg of **XY12**, the average survival time and survival rate of mice increased significantly. However, **XY2** did not show any protective efficacy against *C. albicans* infection at the dose of 10 mg/kg. When the dose reduced to 5 mg/kg, the protective effect of **XY12** was lost. However, increasing the dose of **XY12** not only failed to improve the protective effect, but also showed toxicity in vivo. The survival time of mice injected with 20 mg/kg of **XY12** was shorter than that of mice injected with the solvent (Figure 4). Although **XY12** showed moderate protective effect in vivo, it also caused abdominal irritation in mice, which was reflected by the writhing responses. Therefore, further structural modification of **XY12** was required to enhance its antifungal effect and reduce its irritation to mammals.

## 3. Materials and Methods

### 3.1. Chemistry

#### 3.1.1. General Information

All the starting materials were commercially available reagents and used without further purification. ^1^H, ^19^F and ^13^C-NMR spectra were recorded at 500 MHz for ^1^H, at 126 MHz for ^13^C and at 282 MHz for ^19^F in CDCl_3_ with the BRUKER DRX-500 or 300M NMR spectrometer (Bruker, Billerica, MA, USA). Coupling constants were given in hertz (Hz). GC-MS was conducted using the Agilent MSD Trap XCT (for ESI) and Q-Tof (for HR-ESI-MS) (Agilent Technologies, Santa Clara, CA, USA) and were recorded on the Shimadzu/QP2010 Plus (Kyoto, Japan). All reactions were monitored by thin-layer chromatography (TLC) using silica-gel plates (silica gel 60 F254 0.25 mm).

#### 3.1.2. Synthesis

This was the general procedure for the synthesis of **XY3-17**, **19**: To a mixture of **XY1A** or **XY1B** (0.168 mmol) and **A1-16** (0.202 mmol), acetic anhydride (1.0 mL) was added under an argon atmosphere. The mixture was allowed to stir at 130 °C for 3 h before it was cooled to room temperature and a precipitate formed. The precipitate was washed with diethyl ether (3 × 20 mL), which can be directly used or purified by flash column chromatography for purification to provide **XY3-17** (yield: 55–75%) and **XY19** (yield: 83%) as yellow or dark blue powder. The purity was identified by ^1^H-NMR.

This was the general procedure for the synthesis of **XY2****0**: To a stirred solution of **S1** (0.176 mmol) in DMF, NaH (0.264 mmol) at 0 °C was added under an argon atmosphere. The mixture was allowed to stir at that temperature for 30 min before a solution of **A11** (0.211 mmol) in DMF was added. The mixture was allowed to stir at room temperature for 24 h before it was quenched with saturated aq. NaHCO_3_ was extracted with EtOAc (3 times). The combined organic phases were washed with brine, dried over anhydrous Na_2_SO_4_ and filtered. The solvent was evaporated under vacuum, and the residue was subjected to flash column chromatography for purification using petroleum ether/EtOAc (10%) as eluent to give **XY2****0** (59.8 mg, yield: 94%) as a white powder.

**XY3:** An orange powder. Yield 71% (purity > 95%). ^1^H-NMR (500 MHz, CDCl_3_): *δ* = 8.42 (d, *J* = 16.1 Hz, 1 H), 7.92 (d, *J* = 8.8 Hz, 2 H), 7.85 (s, 2 H), 7.37 (d, *J* = 15.7 Hz, 1 H), 6.90 (d, *J* = 8.7, 2 H), 3.83 (d, *J* = 2.7 Hz, 3 H), 1.48 (d, *J* = 1.9 Hz, 18 H) ppm; ^19^F-NMR (282 MHz, CDCl_3_) *δ* = −78.7 ppm; ^13^C-NMR (126 MHz, CDCl_3_): *δ* = 183.0, 165.2, 164.4, 152.8, 133.7, 127.8, 120.8, 115.2, 113.2, 55.8, 38.6, 28.2 ppm. HRMS (*m*/*z*): [M]^+^ calcd for C_22_H_29_O_2_^+^ 325.2162, found 325.2166; [M]^−^ calcd for CF_3_SO_3_^−^,148.9526, found 148.9529.

**XY4:** An orange powder. Yield 59% (purity > 95%). ^1^H-NMR (500 MHz, CDCl_3_): *δ* = 9.36 (s, 1 H) 8.05 (d, *J* = 15.6 Hz, 1 H), 7.62–7.52 (m, 4 H), 7.05 (d, *J* = 15.7 Hz, 1 H), 6.88 (d, *J* = 8.3 Hz, 2 H), 1.48 (s, 18 H) ppm; ^19^F-NMR (282 MHz, CDCl_3_) *δ* = −78.7 ppm; ^13^C-NMR (126 MHz, CDCl_3_): *δ* = 183.5, 164.7, 162.3, 152.3, 133.5, 126.8, 119.5, 117.2, 112.4, 38.6, 28.0 ppm. HRMS (*m*/*z*): [M]^+^ calcd for C_21_H_2__7_O_2_^+^ 311.2006, found 311.2012; [M]^−^ calcd for CF_3_SO_3_^−^,148.9526, found 148.9528.

**XY6**: A brown powder. Yield 68% (purity > 95%). ^1^H-NMR (500 MHz, CDCl_3_): *δ* = 8.40 (d, *J* = 16.1 Hz, 1 H), 7.98 (dd, *J* = 8.9, 5.4 Hz, 2 H), 7.95 (s, 2 H), 7.44 (d, *J* = 16.0 Hz, 1 H), 7.08–7.02 (m, 2 H), 1.52 (s, 18 H) ppm; ^19^F-NMR (282 MHz, CDCl_3_) *δ* = −78.7, −113.8 ppm; ^13^C-NMR (126 MHz, CDCl_3_): *δ* = 184.4, 165.7, 165.5 (d, *J* = 256.5 Hz), 151.0, 133.5 (d, *J* = 9.1 Hz), 131.10 (d, *J* = 2.9 Hz), 123.13 (d, *J* = 2.0 Hz), 116.70 (d, *J* = 21.9 Hz), 114.3, 38.9, 28.3 ppm. HRMS (*m*/*z*): [M]^+^ calcd for C_21_H_2__6_FO^+^ 313.1962, found 313.1972; [M]^−^ calcd for CF_3_SO_3_^−^,148.9526, found 148.9527.

**XY7:** A brown powder. Yield 64% (purity > 95%). ^1^H-NMR (500 MHz, CDCl_3_): *δ* = 8.32 (d, *J* = 16.1 Hz, 1 H), 8.10 (s, 2 H), 8.00 (d, *J* = 8.0 Hz, 2 H), 7.61 (d, *J* = 8.2 Hz, 2 H), 7.58 (d, *J* = 16.2 Hz, 1 H), 1.54 (s, 18 H) ppm; ^19^F-NMR (282 MHz, CDCl_3_) *δ* = −78.7 ppm; ^13^C-NMR (126 MHz, CDCl_3_): *δ* = 186.0, 165.2, 148.2, 138.5, 132.9, 130.8, 126.2, 118.3, 115.4, 114.7, 39.2, 28.4. HRMS (*m*/*z*): [M]^+^ calcd for C_22_H_2__6_NO^+^ 320.2009, found 320.2015; [M]^−^ calcd for CF_3_SO_3_^−^,148.9526, found 148.9530.

**XY8:** A brown powder. Yield 67% (purity > 95%). ^1^H-NMR (500 MHz, CDCl_3_): *δ* = 8.15 (d, *J* = 16.2 Hz, 1 H), 8.05 (d, *J* = 8.3 Hz, 2 H), 8.00 (s, 2 H), 7.78 (d, *J* = 8.1 Hz, 2 H), 7.37 (d, *J* = 16.2 Hz, 1 H), 2.96 (s, 3 H), 1.56 (s, 18 H) ppm; ^19^F-NMR (282 MHz, CDCl_3_) *δ* = −78.7 ppm; ^13^C-NMR (126 MHz, CDCl_3_): *δ* = 186.0, 164.7, 147.1, 142.1, 139.3, 131.3, 128.0, 126.0, 115.3, 44.3, 39.2, 28.3 ppm. HRMS (*m*/*z*): [M]^+^ calcd for C_22_H_29_O_3_S^+^ 373.1832, found 373.1839; [M]^−^ calcd for CF_3_SO_3_^−^,148.9526, found 148.9527.

**XY9:** A yellow powder. Yield 72% (purity > 90%). ^1^H-NMR (500 MHz, CDCl_3_): *δ* = 8.41 (d, *J* = 16.1 Hz, 1 H), 8.05 (s, 2 H), 7.99 (d, *J* = 8.2 Hz, 2 H), 7.96 (d, *J* = 8.4 Hz, 2 H), 7.63 (d, *J* = 16.0 Hz, 1 H), 3.91 (s, 3 H), 1.51 (s, 18 H) ppm; ^19^F-NMR (282 MHz, CDCl_3_) *δ* = −78.7 ppm; ^13^C-NMR (126 MHz, CDCl_3_): *δ* = 185.3, 166.3, 165.5, 149.9, 138.5, 133.0, 130.4, 130.3, 125.6, 115.0, 52.5, 39.1, 28.3 ppm. HRMS (*m*/*z*): [M]^+^ calcd for C_23_H_29_O_3_^+^ 353.2111, found 353.2118; [M]^−^ calcd for CF_3_SO_3_^−^,148.9526, found 148.9528.

**XY10:** A dark blue powder. Yield 59% (purity > 95%). ^1^H-NMR (500 MHz, CDCl_3_): *δ* = 8.23 (d, *J* = 15.0 Hz, 1 H), 7.85 (d, *J* = 8.7 Hz, 2 H), 7.37 (s, 2 H), 7.06 (d, *J* = 15.1 Hz, 1 H), 6.74 (d, *J* = 9.0 Hz, 2 H), 3.50 (q, *J* = 7.1 Hz, 4 H), 1.42 (s, 18 H), 1.24 (t, *J* = 7.1 Hz, 6 H) ppm; ^19^F-NMR (282 MHz, CDCl_3_) *δ* = −78.7 ppm; ^13^C-NMR (126 MHz, CDCl_3_): *δ* = 178.5, 160.7, 153.6, 153.2, 123.4, 116.1, 112.8, 45.5, 37.8, 28.2, 12.9 ppm. HRMS (*m*/*z*): [M]^+^ calcd for C_25_H_3__1_NO^+^ 366.2791, found 366.2793; [M]^−^ calcd for CF_3_SO_3_^−^,148.9526, found 148.9530.

**XY11:** A dark blue powder. Yield 57% (purity > 92%). ^1^H-NMR (500 MHz, CDCl_3_): *δ* = 8.35 (d, *J* = 15.4 Hz, 1 H), 7.78 (d, *J* = 8.7 Hz, 2 H), 7.68 (s, 2 H), 7.37 (m, 4 H), 7.24–7.17 (m, 7 H), 6.96 (d, *J* = 8.5 Hz, 2 H), 1.48 (s, 18 H) ppm; ^19^F-NMR (282 MHz, CDCl_3_) *δ* = −78.7 ppm; ^13^C-NMR (126 MHz, CDCl_3_): *δ* = 181.3, 163.5, 153.7, 152.9, 145.4, 133.8, 130.0, 127.0, 126.9, 126.2, 122.4, 119.8, 119.4, 119.1, 112.1, 38.4, 28.3 ppm. HRMS (*m*/*z*): [M]^+^ calcd for C_33_H_3__6_NO^+^ 462.2791, found 462.2795; [M]^−^ calcd for CF_3_SO_3_^−^,148.9526, found 148.9526.

**XY12:** A dark blue powder. Yield 75% (purity > 95%). ^1^H-NMR (500 MHz, CDCl_3_): *δ* = 8.23 (d, *J* = 15.0 Hz, 1 H), 7.84 (d, *J* = 8.5 Hz, 2 H), 7.35 (s, 2 H), 7.06 (d, *J* = 15.1 Hz, 1 H), 6.64 (d, *J* = 8.7 Hz, 2 H), 3.50–3.47 (m, 4 H), 2.10–2.04 (m, 4 H), 1.42 (s, 18 H) ppm; ^19^F-NMR (282 MHz, CDCl_3_) *δ* = −78.6 ppm; ^13^C-NMR (126 MHz, CDCl_3_): *δ* = 178.5, 160.6, 153.6, 153.1, 123.7, 122.2, 119.7, 116.1, 113.7, 48.5, 37.9, 28.2, 25.4 ppm. HRMS (*m*/*z*): [M]^+^ calcd for C_25_H_3__4_NO^+^ 364.2635, found 364.2644; [M]^−^ calcd for CF_3_SO_3_^−^,148.9526, found 148.9529.

**XY 13:** A dark blue powder. Yield 61% (purity > 95%). ^1^H-NMR (500 MHz, CDCl_3_): *δ* = 8.27 (d, *J* = 15.1 Hz, 1 H), 7.86 (d, *J* = 8.8 Hz, 2 H), 7.45 (s, 2 H), 7.13 (d, *J* = 15.1 Hz, 1 H), 6.90 (d, *J* = 8.7 Hz, 2 H), 3.56 (m, 4 H), 1.73–1.69 (m, 6 H), 1.44 (s, 18 H) ppm; ^19^F-NMR (282 MHz, CDCl_3_) *δ* = −78.7 ppm; ^13^C-NMR (126 MHz, CDCl_3_): *δ* = 183.7, 164.0, 149.4, 133.0, 122.7, 121.7, 119.5, 118.9, 113.6, 54.8, 38.8, 28.2, 24.1, 22.1 ppm. HRMS (*m*/*z*): [M]^+^ calcd for C_26_H_3__6_NO^+^ 378.2791, found 378.2792; [M]^−^ calcd for CF_3_SO_3_^−^,148.9526, found 148.9529.

**XY14:** A dark blue powder. Yield 63% (purity > 95%). ^1^H-NMR (500 MHz, CDCl_3_): *δ* = 8.22 (d, *J* = 15.2 Hz, 1 H), 7.81 (d, *J* = 8.4 Hz, 2 H), 7.37 (s, 2 H), 7.05 (d, *J* = 15.1 Hz, 1 H), 6.37 (d, *J* = 8.6 Hz, 2 H), 4.14 (t, *J* = 7.5 Hz, 4 H), 2.48 (m, 2 H), 1.42 (s, 18 H) ppm; ^19^F-NMR (282 MHz, CDCl_3_) *δ* = −78.7 ppm; ^13^C-NMR (126 MHz, CDCl_3_): *δ* = 178.7, 160.8, 155.0, 153.6, 123.9, 116.2, 111.2, 51.5, 37.9, 28.2, 16.2 ppm. HRMS (*m*/*z*): [M]^+^ calcd for C_24_H_3__2_NO^+^ 350.2478, found 350.2484; [M]^−^ calcd for CF_3_SO_3_^−^,148.9526, found 148.9530.

**XY15:** A dark blue powder. Yield 67% (purity > 90%). ^1^H-NMR (500 MHz, CDCl_3_): *δ* = 8.32 (d, *J* = 15.2 Hz, 1 H), 7.89 (d, *J* = 8.5 Hz, 2 H), 7.60 (s, 2 H), 7.24 (d, *J* = 16.3 Hz, 1 H), 6.92–6.86 (m, 1 H), 3.88–3.75 (m, 1 H), 3.5–3.39 (m, 1 H), 1.45 (s, 18 H) ppm; ^19^F-NMR (282 MHz, CDCl_3_) *δ* = −78.7 ppm; ^13^C-NMR (126 MHz, CDCl_3_): *δ* = 180.7, 163.0, 154.9, 153.2, 134.7, 125.2, 118.3, 114.0, 111.5, 66.5, 46.9, 38.2, 28.2 ppm. HRMS (*m*/*z*): [M]^+^ calcd for C_25_H_3__4_NO_2_^+^ 380.2584, found 380.2589; [M]^−^ calcd for CF_3_SO_3_^−^,148.9526, found 148.9523.

**XY16:** A dark blue powder. Yield 63% (purity > 92%). ^1^H-NMR (500 MHz, CDCl_3_): *δ* = 8.30 (d, *J* = 15.0 Hz, 1 H), 7.87 (d, *J* = 8.5 Hz, 2 H), 7.47 (s, 2 H), 7.14 (d, *J* = 15.0 Hz, 1 H), 6.59 (d, *J* = 8.3 Hz, 2 H), 5.42 (d, *J* = 52.0 Hz, 1 H), 3.72–3.54 (m, 4 H), 2.49–2.35 (m, 1 H), 2.29–2.11 (m, 1 H), 1.43 (s, 18 H) ppm; ^19^F-NMR (282 MHz, CDCl_3_) *δ* = −78.7,−173.5–−179.0 (m) ppm; ^13^C-NMR (126 MHz, CDCl_3_): *δ* = 179.3, 161.7, 153.7, 152.5, 124.1, 122.3, 119.8, 116.8, 113.5, 92.39 (d, *J* = 176.4 Hz), 54.80 (d, *J* = 23.1 Hz), 46.0, 38.0, 32.05 (d, *J* = 21.8 Hz), 28.2 ppm. HRMS (*m*/*z*): [M]^+^ calcd for C_25_H_3__3_FNO^+^ 382.2541, found 382.2544; [M]^−^ calcd for CF_3_SO_3_^−^,148.9526, found 148.9523.

**XY17:** A dark blue powder. Yield 55% (purity > 92%). ^1^H-NMR (500 MHz, CDCl_3_): *δ* = 8.33 (d, *J* = 15.0 Hz, 1 H), 7.90 (d, *J* = 8.5 Hz, 2 H), 7.49 (s, 2 H), 7.16 (d, *J* = 15.0 Hz, 1 H), 6.65 (d, *J* = 8.4 Hz, 2 H), 5.46 (d, *J* = 4.0 Hz, 1 H), 3.75 (dd, *J* = 12.5, 4.7 Hz, 1 H), 3.67–3.59 (m, 2 H), 3.56 (d, *J* = 12.5 Hz, 1 H), 2.30–2.25 (m, 2 H), 1.45 (s, 18 H) ppm; ^19^F-NMR (282 MHz, CDCl_3_) *δ* = −78.6 ppm; ^13^C-NMR (126 MHz, CDCl_3_): *δ* = 179.4, 161.7, 153.8, 152.5, 124.1, 122.3, 119.7, 116.9, 113.5, 73.2, 54.0, 46.3, 38.0, 31.1, 28.3 ppm. HRMS (*m*/*z*): [M]^+^ calcd for C_25_H_3__4_NO_2_^+^ 380.2584, found 380.2590; [M]^−^ calcd for CF_3_SO_3_^−^,148.9526, found 148.9528.

**XY19:** A dark blue powder. Yield 83% (purity > 95%). ^1^H-NMR (500 MHz, CDCl_3_): *δ* = 8.31 (s, 2 H), 8.25 (d, *J* = 15.8 Hz, 1 H), 8.20 (d, *J* = 7.7 Hz, 4H), 7.94 (d, *J* = 8.2 Hz, 2 H), 7.82–7.78 (m, 2 H), 7.75–7.66 (m, 4 H), 7.53 (d, *J* = 8.1 Hz, 2 H), 7.47 (d, *J* = 15.8 Hz, 1 H), 3.87–3.76 (m, 4 H), 2.37–2.29 (m, 4 H) ppm; ^19^F-NMR (282 MHz, CDCl_3_) *δ* = −78.6 ppm; ^13^C-NMR (126 MHz, CDCl_3_): *δ* = 169.6, 160.4, 147.9, 144.6, 135.7, 132.6, 130.5, 128.7, 128.0, 123.6, 121.0, 118.6, 113.8, 57.9, 24.3 ppm. HRMS (*m*/*z*): [M]^+^ calcd for C_29_H_26_NO^+^ 404.2009, found 404.2014; [M]^−^ calcd for CF_3_SO_3_^−^,148.9526, found 148.9528.

**XY2****0:** A white powder. Yield 94% (purity > 95%). ^1^H-NMR (300 MHz, CDCl_3_): *δ* = 7.44 (d, *J* = 8.6 Hz, 2 H), 7.36–7.33 (m, 2 H), 7.31 (m, 1 H), 7.06 (d, *J* = 16.2 Hz, 1 H), 6.95 (d, *J* = 16.2 Hz, 1 H), 6.59 (d, *J* = 8.2 Hz, 2 H), 3.45–3.27 (m, 4 H), 2.18–1.91 (m, 4 H), 1.38 (s, 18 H) ppm; ^13^C-NMR (75 MHz, CDCl_3_): *δ* = 151.0, 147.5, 137.5, 128.5, 127.8, 125.0, 121.2, 120.5, 112.0, 47.8, 35.0, 31.6, 25.6 ppm. HRMS (*m*/*z*): [M + H]^+^ calcd for C_25_H_3__5_N^+^ 362.2842, found 362.2841.

### 3.2. In Vitro Antifungal Activity

The standard *C. albicans* SC5314 was donated by Professor *William A. Fonzi* of Georgetown University in the United States. Clinical isolates of *C. albicans*, *C. glabrata*, *C. tropicalis*, *C. krusei* and *C. parapsilosis* were isolated from patients of Changhai Hospital and identified by biochemistry and morphology. The antifungal minimum inhibitory concentration (MIC) was measured by serial dilution in 96-well plates with RPMI 1640 medium as described in the Clinical and Laboratory Standards Institute (CLSI) guidelines (CLSI M27-A3). The MIC is defined as the concentration that reduced fungal growth by more than 95%. All compounds were dissolved in DMSO. The OD_600_ was determined after *Candida* was cultured at 30 °C for 48 h. The MIC of each compound was calculated by OD_600_. All the antifungal activity was detected at least two times. The in vitro antifungal activities of **XY1A****-20** were evaluated by the *C. albicans* standard isolate SC5314.

### 3.3. Cytotoxicity against Human Umbilical Vein Endothelial Cells (HUVECs)

HUVECs were cultured to detect the cytotoxicity of **XY12** and **XY2** as described. HUVECs were cultured in DMEM containing 10% fetal bovine serum (FBS). Then, a 100 μL suspension of HUVECs (1 × 10^5^ cells/mL) was added to 96-well tissue culture plates and incubated at 37 °C for 3 h. After incubation, the supernatant was replaced by 100 μL of fresh DMEM complete mediums containing different concentrations of **XY12** and **XY2**. After incubation for 24 h, the supernatant was added with 10 μL of CCK-8 agents and cultured at 37 °C for 2 h. Finally, cell viability was assessed by measuring OD_450_ [18].

### 3.4. In Vivo Antifungal Activity

8-week-old female ICR mice were purchased from Shanghai SLAC Laboratory Animal Co., Ltd (Shanghai, China). The mice were injected with a 200 μL PBS suspension of *C. albicans* SC5314 (7.5 × 10^6^ CFU/mL) through the tail vein. **XY12** and **XY2** compounds were solved in the DPH buffer (5% DMSO, 30% PEG 400 and 65% ddH_2_O) and injected intraperitoneally 2 h after infection. **XY12** and **XY2** were injected in the mice once a day for 5 days. After that, the mice were observed daily for survival.

## 4. Conclusions

In summary, based on the lead compound **SM21** (**XY2**), we synthesized a series of pyrylium salt derivatives and evaluated their antifungal activities. Most of the designed compounds exhibited moderate in vitro antifungal activities against *C. albicans*. Among them, the most promising antifungal agent was coded as **XY12**. Compared with the lead compound **XY2**, **XY12** showed lower cytotoxicity and higher activity against *Candida* species, including the fluconazole-resistant *C. albicans*. Additionally, the in vivo antifungal activity of **XY12** was also enhanced. The SARs study demonstrated that the electron-donating level of terminal groups on the benzene ring determined antifungal activity. At the same time, further activity and selectivity strongly depended on the ring size and steric hindrance of terminal cyclic alkylamine. The character of lipophilic cations responsible for activity implies that this type of pyrylium dye exhibited the antifungal activity, probably via targeting mitochondria. Researches on the antifungal mechanisms of pyrylium salts will provide more information for further structural modifications.

## Data Availability

Not applicable.

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
