# Peer review of "Design, Synthesis and Antifungal Evaluation of Novel Pyrylium Salt In Vitro and In Vivo"

_molecules, 2022, doi:10.3390/molecules27144450_

Round 1

Reviewer 1 Report

The manuscript “Design, Synthesis and Antifungal Evaluation of Novel Pyrylium Salt in vitro and in vivo” by authors Yue Zhang et al contains the data on synthesis and biological evaluation of pyrylium salts. Authors investigated the structure-activity relationships between the known pyrylium salt SM21 and its analogs. It was found that compound XY14 has the promising anti-Candida activity.

Some remarks:

1. Almost all compounds have CF3SOO anion. Authors should give 19F NMR spectra for the new compounds. The 13C NMR spectrum for compound XY11 should be enhanced (very low intensity of signals). CF3 group in 13C NMR must has the quartet signal with J 270-280 Hz. I did not find this information in Experimental part. 

2. To my mind, it is no sufficient the HRMS data for salts because only the peak of cation can be observed. Please, provide the elemental analysis for the new compounds or HRMS data for anion form.

3. The MIC should be converted in μM to minimize the influence of molecular mass of compounds on the bioactivity results.

4. Please, provide the data on acute toxicity in vivo for the leading compounds.

 Given all the above, I recommend to accept the manuscript after minor revision.

Author Response

Dear Reviewer,

Thank you very much for your time involved in reviewing the manuscript and your very encouraging comments on the merits.

Point 1: Almost all compounds have CF3SO3 anion. Authors should give 19F NMR spectra for the new compounds. The 13C NMR spectrum for compound XY11 should be enhanced (very low intensity of signals). CF3 group in 13C NMR must has the quartet signal with J 270-280 Hz. I did not find this information in Experimental part. 

Response 1: In the revised manuscript, we added the 19F NMR spectra. The quartet signal of CF3 group in 13C NMR was the co-solvent TFA’ signal, which was used to improve the solubility reported in references. The CF3SO3’ signal can not be found in the 13C NMR spectra for their low solubility. The triflate anion was ascertained by HRMS and 19F NMR spectra. Although we have try our best to purified compound XY11, the purity was only 70%, so we have to delete this compound.

Point 2: To my mind, it is no sufficient the HRMS data for salts because only the peak of cation can be observed. Please, provide the elemental analysis for the new compounds or HRMS data for anion form.

Response 2: In the revised manuscript, we added the HRMS data both cation and anion form.

Point 3: The MIC should be converted in μM to minimize the influence of molecular mass of compounds on the bioactivity results.

Response 3: In the revised manuscript, we have converted the unit of MIC value ug into μM.

Point 4: Please, provide the data on acute toxicity in vivo for the leading compounds.

Response 4: In our study, XY14 (XY12 in the revised manuscript) still has the problems of toxicity and irritation. In the revised manuscript, we added the survival rate of C. albicans-infected mice treated with high doses of XY14 (XY12 in the revised manuscript)  (Figure 4). Compared with mice injected with solvent, the survival time of mice injected with 20 mg/kg or 30 mg/kg of XY14 (XY12 in the revised manuscript)  was significantly shorter, which also reflected the high toxicity of XY14 (XY12 in the revised manuscript). In the future, we will synthesize less toxic compounds based on the XY14 (XY12 in the revised manuscript) structure, and then carry out the acute toxicity studies.

Reviewer 2 Report

This manuscript “Design, Synthesis and Antifungal Evaluation of Novel Pyrylium Salt in vitro and in vivo” by Liu, Wang and co-workers described an interesting activity of Pyrylium Salts as an antifungal agent. The authors described that XY14 showed improved antifungal activity and lower cytotoxicity with respect to the lead compound XY2 against Candida species. The manuscript was written well and organized. But I will support publishing the revised manuscript if the authors clarify the following points-

1. All the synthesized compounds contain triflate as the counter anion. But the 19F-NMR data were not recorded. I will suggest recording all the 19F-NMR.

2. The purity and the yields for the individual compounds have not been mentioned.

3. Repurify XY7, XY8, XY9, XY10, XY11, XY15, XY16, XY19

4. In the 1H NMR of XY4, the peak around 9.3 ppm has not been integrated.

5. In the 13C of XY6, the peak around 162 (as a doublet with JC-F around 250 Hz) has been ignored. The peak is for the carbon attached to the fluorine atom.

6. In the 1H NMR of XY12, the peak of 7.37 ppm has been mentioned as a triplet, how it can be possible that peaks in the aromatic region to be a triplet?

7. The spectral data of XY13 is missing.

8. For the 1H NMR of XY14, reintegrate all the peaks. I will suggest recording the 1H NMR with more scans.

9. For the 13C of XY18, please calculate and mention the JC-F values for the peaks coming as doublets.

10. The 13C NMR of XY21 contains extra peaks.

11. Rerecord the 13C NMR of XY22.

12. The authors have tested a series of cyclic alkylamines like pyrrolidine, morpholine etc. It will be very interesting if they may synthesize and include the piperazine substituted derivative in this manuscript.

13. Do the authors tried to test the activity with the compound having other counter anions such as perchlorate instead of triflate?

Author Response

Dear Reviewer,

Thank you very much for your time involved in reviewing the manuscript and your very encouraging comments on the merits.

Point 1: All the synthesized compounds contain triflate as the counter anion. But the 19F-NMR data were not recorded. I will suggest recording all the 19F-NMR.

Response 1: Thanks for your suggestions. In the revised manuscript, we added the 19F NMR spectra.

Point 2: The purity and the yields for the individual compounds have not been mentioned.

Response 2: Thanks for your suggestions. In the revised manuscript, we added the yield and the purity identfied by NMR.

Point 3: Repurify XY7, XY8, XY9, XY10, XY11, XY15, XY16, XY19

Response 3: Thanks for your suggestions. We have tried our best to repurify these compounds, but the purity of XY11 was only 70%, so we have to delete this compound and its other data.

Point 4: In the 1H NMR of XY4, the peak around 9.3 ppm has not been integrated.

Response 4: Thanks for your suggestions. Compound XY4 have low solubility, so we added TFA as co-solvents, and the peak around 9.3 ppm covered the signal of phenol, in the revised manuscript, we integrated the phenol signal partly.

Point 5: In the 13C of XY6, the peak around 162 (as a doublet with JC-F around 250 Hz) has been ignored. The peak is for the carbon attached to the fluorine atom.

Response 5: Thanks for your suggestions. In the revised manuscript, we pick all the signal of compound XY6.

Point 6: In the 1H NMR of XY12, the peak of 7.37 ppm has been mentioned as a triplet, how it can be possible that peaks in the aromatic region to be a triplet?

Response 6: Thanks for your suggestions. In the revised manuscript, we revised the peak of 7.37 ppm as multiplet.

Point 7: The spectral data of XY13 is missing.

Response 7: Thanks for your suggestions. Although we have try our best to purified compound XY13, the purity was still below 70%, so we have to delete this compound and its other data.

Point 8: For the 1H NMR of XY14, reintegrate all the peaks. I will suggest recording the 1H NMR with more scans.

Response 8: Thanks for your suggestions. we rerecord the 1H NMR of XY14 (XY12 in the revised manuscript), and we also change the deuterated solvent as DMSO-d6, howere the baseline was still not very smooth, so we reprocessed the before one in revised manuscript.

Point 9: For the 13C of XY18, please calculate and mention the JC-F values for the peaks coming as doublets.

Response 9: Thanks for your suggestions. In the revised manuscript, we added the  JC-F values of XY18 (XY16 in the revised manuscript) and XY6.

Point 10: The 13C NMR of XY21 contains extra peaks.

Response 10: Thanks for your suggestions. To improve the solubility, TFA as co-solvent was added, and the extra peaks of XY21 (XY19 in the revised manuscript) were the signal of TFA (160.43 (q, J = 42.6 Hz), 114.64 (q, J = 285.3 Hz)).

Point 11: Rerecord the 13C NMR of XY22.

Response 11: Thanks for your suggestions. In the revised manuscript, we reracorded the 13C NMR of XY22 (XY20 in the revised manuscript).

Point 12: The authors have tested a series of cyclic alkylamines like pyrrolidine, morpholine etc. It will be very interesting if they may synthesize and include the piperazine substituted derivative in this manuscript.

Response 12: Thanks for your suggestions. This interesting design is in our plan, and we also want to obtain the imidazole and other five-membered heterocyclic rings, unfortunately these derivatives not easy to access by same method. But according to our structure-activity relationship study, we found substitute on the cyclic is not good for the antifugual acitvity (XY13 and XY15 in our revised manuscript).

Point 13: Do the authors tried to test the activity with the compound having other counter anions such as perchlorate instead of triflate?

Response 13: Thanks for your suggestions. Currently, we have not tested the effect of anions on the biological activity of pyrylium salts, which was mainly due to the following reasons. At first, the antifungal activity of XY2 (triflate) was the same as the commercially available SM21 (perchorate salt) reported by Wong et al. Secondly, the RPMI1640 medium and PBS buffers used in our experiments contains different anions such as bicarbonate, chloride and nitrate, which exclude the effect of some anions on the activity of pyrylium salts. Thirdly, our structure-activity relationship study showed that the cationic part of pyrylium significantly affected its antifungal activity, so we believe that the cationic is critical. This study mainly focuses on the influence of pyrylium salt cations and we will further explore the influence of anions in the future.

Wong SSW, Kao RYT, Yuen KY, Wang Y, Yang D, Samaranayake LP, et al. (2014) In Vitro and In Vivo Activity of a Novel Antifungal Small Molecule against Candida Infections. PLoS ONE 9(1): e85836. https://doi.org/10.1371/journal.pone.0085836

Reviewer 3 Report

This work describes the synthesis and in vitro/in vivo evaluation of new pyrylum salts against Candida spp. The results are very interesting and promissing, but some corrections are necessary before publication:

Minor revisions:

1. Page 2, line 69: replace "Figure 1" with "Schemes 1, 2 and 3".

2. Page 4, line 89: replace "antibacterial" with "antimicrobial".

3. Page 4, line 94: replace "XY1" with "XY1B".

Major revisions:

1. In Results and Discussion section, include a discussion of the chemical characterization methods used, main signs, multiplicity, coupling constant values of the obtained compounds. Include also infrared data of the synthesized substances, as well as their spectra in Supplementary Material Section.

2. Include all high resolution mass spectra of the synthesized compounds in Supplementary Material section. 

3. In some 1H and 13C NMR spectra of the synthesized compounds there are missing or leftover signals. I suggest checking all spectra and reprocessing/redoing if necessary.

Author Response

Dear Reviewer,

Thank you very much for your time involved in reviewing the manuscript and your very encouraging comments on the merits.

Point 1: Page 2, line 69: replace "Figure 1" with "Schemes 1, 2 and 3".

Response 1: Thanks for your suggestions. xIn the revised manuscript, “In our studies, a series of pyrylium salts were rationally designed and synthesized base on the chemical space investigation of XY2 (Figure 1).” was replaced to “In our studies, a series of pyrylium salts were rationally designed and synthesized base on the chemical space investigation of XY2 (Schemes 1, 2 and 3).”

Point 2: Page 4, line 89: replace "antibacterial" with "antimicrobial".

Response 2: Thanks for your suggestions. In the revised manuscript, “The main purpose of this study is to explore the chemical space and analysis the structure-activity relationship of pyrylium salt derivatives against fungi, so as to find novel skeleton antimicrobial agents with high efficiency and low toxicity.” Was revised as “The main purpose of this study is to explore the chemical space and analysis the structure-activity relationship of pyrylium salt derivatives against fungi, so as to find novel skeleton antibacterial agents with high efficiency and low toxicity.”

Point 3: Page 4, line 94: replace "XY1" with "XY1B"

Response 3: Thanks for your suggestions. In the revised manuscript, “Pyrylium salt (XY1A) and N,N-dimethylaniline (XY1) showed no significant activity (MIC > 64 μg/mL), which revealed that both fragments motif were required for antifungal activity.” Was revised as “Pyrylium salt (XY1A) and 4-(Dimethylamino)benzaldehyde (A1)  showed no significant activity (MIC > 179.7 μM), which revealed that both fragments motif were required for antifungal activity.”

Point 4: In Results and Discussion section, include a discussion of the chemical characterization methods used, main signs, multiplicity, coupling constant values of the obtained compounds. Include also infrared data of the synthesized substances, as well as their spectra in Supplementary Material Section.

Response 4: Thanks for your suggestions. In the revised manuscript, we added the discussion of the chemical characterization of this pyrylium salts. For our experiment conditions, we have not recorded the infrared data, we added the 19F NMR data to improve certainty of chemical characterization.

Point 5: Include all high resolution mass spectra of the synthesized compounds in Supplementary Material section.

Response 5: Thanks for your suggestions. In the revised manuscript and supplementary material, we added the high resolution mass spectra.

Point 6: In some 1H and 13C NMR spectra of the synthesized compounds there are missing or leftover signals. I suggest checking all spectra and reprocessing/redoing if necessary.

Response 6: Thanks for your suggestions. In the revised manuscript and supplementary material, we have checked all the spectra, and reprocessed all the 1H and 13C NMR spectra, we replaced the 13C NMR spectra of XY-22 (XY20 in the revised manuscript), deleted the data of compounds XY-11, which purity was only 50%, although we have try our best to purify it.

Round 2

Reviewer 2 Report

I highly recommend this revised manuscript for the "Molecules".